# The role of Mechanistic Interpretability in 'unveiling' the emergent representations of Large Language Models

## Abstract

Despite the fact that significant progress has been achieved by Large Language Models (LLMs), the internal mechanisms that enable the generalization and reasoning abilities still need to be explored. This gap, nurtured by phenomena such as hallucinations, adversarial perturbations, and misaligned human expectations, leads to suspicions that hinder LLMs' safe and beneficial use. This paper provides a comprehensive overview of the current state of explainability approaches related to investigating the underlying mechanisms of LLMs. Therefore, we explore the strategic components we would expect to lay the foundation for generalization capabilities by studying the means to quantify the knowledge acquired and delivered by LLMs and, in particular, discerning the composition and encoding of knowledge within parameters by analyzing mechanistic interpretability, probing techniques, and representation engineering. Finally, we use a mechanistic perspective to explain emergent phenomena that arose in training dynamics, best exemplified by memorisation and generalisation.

## 1 Introduction

Large Language Models (LLMs) such as GPT-s OpenAI (2023), Claude-s AnthropicAI (2023), Gemini-s Team et al. (2023) and Llama-sTouvron et al. (2023) have been highly influential across a diverse range of language understanding and reasoning tasks. Although their performances have been thoroughly investigated across various benchmarks, the principles, properties, and explanations behind their generalization process still need to be better understood. Their black-box nature, reinforced by the commercialization, leading to strategic secrecy about model architectures and training details, does not contribute to interpretability and boosts the difficulty of interpretation Dalal et al. (2024). Finally, these models are susceptible to hallucinations and adversarial perturbations Geirhos et al. (2020); Camburu et al. (2020), often producing plausible but factually incorrect answers Ji et al. (2023); Huang et al. (2023). As the size and complexity of LLM architectures increase, the systematic study of explanation generation becomes crucial to interpret better and validate the LLM's internal inference and reasoning processes Tenney et al. (2019); Lampinen et al. (2022); Wei et al. (2023).

These phenomena have been extensively observed by incorporating explainability-by-design approaches into models Danilevsky et al. (2020b); however, they do not explain the underlying dynamics and cannot track the acquired knowledge that is often transformed by LLMs in ways that fundamentally differ from traditional machine learning approaches Belinkov et al. (2020).

As the need to explore the internal knowledge of LLMs becomes predominant, a crucial technique that has recently emerged is mechanistic interpretability, which allows one to decompose the internal mechanism of the model by identifying and analyzing its core components, such as neurons, hidden layers, or attention mechanisms. This analysis enables the investigation of the model's causal capabilities, helping to make them more transparent, interpretable, and reliable. Mechanistic interpretability has the prospect of addressing some of the current issues in the field:

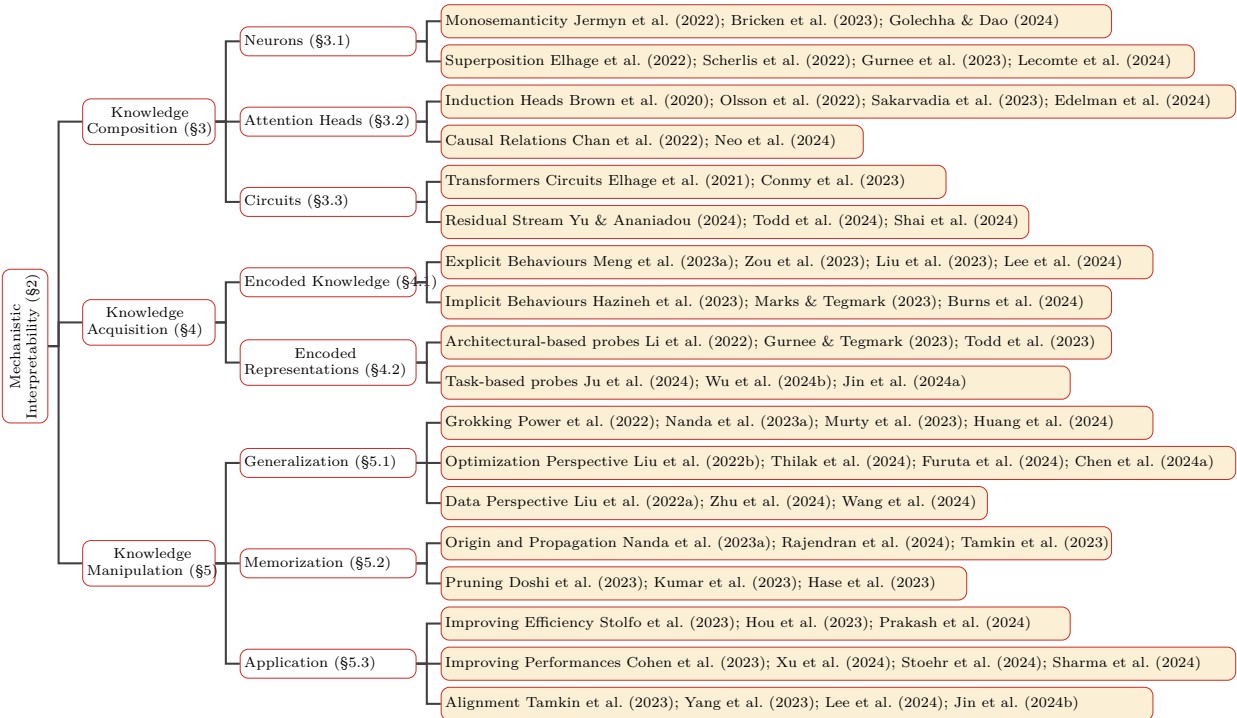

Figure 1: Taxonomy of approaches that use Mechanistic Interpretability viewpoint.

**Structuring knowledge composition** Unraveling the composition of knowledge in model architectures by investigating the components that enable knowledge acquisition;

**Acquiring knowledge encoding** Exploring the content of intermediate representations by discerning the acquired knowledge from the behavioural emergent patterns;

**Emergent abilities** Understanding the mechanisms that lead to generalization in the training process, outlining the boundary within memorization and the surrounding phenomena.

We investigate the following points by providing a systematic overview of the existing literature (reported in Taxonomy in Figure 1) revealing the mechanisms behind the functioning of LLMs by contributing in the following way:

Firstly, we give some preliminaries of the challenges overcome by the mechanistic interpretability approach (§2). Secondly, we proceed to analyze the relevant works that aim to investigate how knowledge is composed in model architectures (§3). Hence, by decomposing the functionality of each model component, we aim to interpret how they operate at the level of neurons, circuits, and heads of attention. Thirdly, we examine how knowledge is encoded internally in intermediate representations (§4) by providing an in-depth overview of representation engineering's pipelines to explain model-specific behaviour. Finally, having collected a series of evidence supported by the insights of the analyses mentioned above, they can play a strategic role in improving the models in terms of superior performance through internal modifications of the most relevant models (§5).

Our contribution complements the existing survey articles on the explainability of LLMs that summarize explainability techniques Danilevsky et al. (2020a) or discuss the importance Saeed & Omlin (2021). We take the next step, following the research line done in Nanda et al. (2023a); Golechha & Dao (2024); Luo & Specia (2024); Ferrando et al. (2024) by reviewing existing studies that focus on discovering the inner workings of LLMs. Complementing previous studies, we identify factors that contribute to the reasoning abilities of LLMs through explainability techniques and control of the training process. Furthermore, we

examine the state-of-the-art knowledge on the internal workings of LLMs and analyze how this knowledge can further improve the inner wisdom of the mechanisms of models and consequently de-opacify the inner workings, benefiting humans. An in-depth understanding of how LLMs work is strategic for the safer use of LLMs in real-world scenarios and critical applications.

## 2 A new paradigm: Mechanistic Interpretability

The mechanistic interpretability paradigm ( known as reverse engineering Krueger (2022)) refers to the activity of investigating Artificial Neural Networks (ANN) to understand the underlying components and mechanisms that determine their behaviour Olah et al. (2020a). In fact, examining the inside of neural networks allows one to perceive the rich internal structures of patterns Geva et al. (2022a). This approach takes the next step from conventional interpretability methods, as discussed in §2.1. Unconventionally, mechanistic interpretation draws on fields, such as neuroscience, to study the connections between individual neurons. Thus, considering each neuron as unique and following its weight, an intricate picture emerges of how neural networks function through interconnected *circuits* that implement meaningful algorithms Conmy et al. (2023); Hanna et al. (2023). This mechanistic view enables the deep analysis of artificial neural systems in that the individual parts, the neurons, play a comprehensible role, and their circuits of connections implement factual relations about the world. Indeed, it is possible to observe the step-by-step construction of abstract concepts, such as circle detectors, animal faces, cars, and logical operations Olah et al. (2020a). A micro-level mechanistic view of LLMs allows for a deeper understanding of their macro-level behaviour. This mechanistic perspective represents a paradigm shift in interpretability, which aims to unpack the causal factors that drive model results.

### 2.1 The role of Mechanistic Interpretability

The mechanistic view of mechanistic interpretability represents a paradigm shift towards a deeper understanding of the dynamics occurring in ANNs Bereska & Gavves (2024); Ferrando et al. (2024). Mechanistic interpretability contributes in the following ways:

- **Implementation Perspective:** Mechanistic interpretability aims to decipher the complexities and mechanisms present in pre-trained models without needing to build models that can be explained explicitly by adapting to existing architectures Friedman et al. (2023).

- **Vision Perspective:** Mechanistic interpretability brings a global view as it aims to comprehend models as a whole through the lens of high-level concepts and circuits. In contrast, traditional explicability approaches focus on explaining specific predictions made by models, e.g., feature attribution techniques, which have a limited view of the whole phenomena Zimmermann et al. (2024).

- **Operation Perspective:** Mechanistic interpretability aligns with white-box analysis, as direct access to a model's internal parameters and activations is required. In contrast to black-box explainability tools (e.g., LIME Ribeiro et al. (2016) and SHAP Lundberg & Lee (2017)), which operate solely based on model inputs and outputs, mechanistic interpretability operates on internal mechanisms by handling unique features.

From what emerges, mechanistic interpretability is a strategic approach that allows for an in-depth understanding of ANNs. It highlights a global and post-hoc standpoint, concentrating on model-specific analysis by interpreting complicated systems' internal mechanisms and intrinsic logic. This vision drives this approach as necessary for boosting transparency and building trust in LLM, particularly in high-risk scenarios where understanding the underlying motivations of ANN models is as paramount as the decisions themselves.

## 3 The Knowledge Composition

One of the strategic points to the success of Large Language Models lies in the complex equilibrium established between extensive training data sets and intricate model architectures Kaplan et al. (2020); Hoffmann

et al. (2022). Although the practice of releasing open-source model specifications is increasing, the exact mechanisms through which these models acquire and process large amounts of knowledge still need to be discovered. In particular, the exploration of the individual contributions of individual model components and their role in the overall function of LLMs still remains as a black-box Das & Rad (2020). Mechanistic Interpretability approaches, bypassing the shortcomings of previous methods, as discussed in §2, are concerned with proposing methods that enable to interpret LLMs at a granular level, such as neurons (§3.1), attention heads (§3.2), and circuits (§3.3).

## 3.1 Neurons

The minimal parts within LLMs are the neurons, which are the fundamental building blocks responsible for encoding knowledge and patterns Golechha & Dao (2024). Similar to non-artificial neural structures, neurons can be activated by several unrelated terms, a phenomenon known as polysemy Olah et al. (2020a); Xu & Poo (2023). This feature poses more significant difficulties in mechanistically understanding how patterns work. However, several emerging works Gurnee et al. (2023); Bricken et al. (2023) have introduced key instruments that have shown two structural points in the formation of neuroular knowledge: *Monosemanticity* (§3.1.1) and *Superposition* (§3.1.2).

### 3.1.1 Monosematicity

In language artificial learning, it is not very easy to distinguish concepts in polysemantic neurons, as a variety of different terms can activate them. In contrast, monosemantic neurons, associated with a single concept, are easier to interpret. For this reason, analyzing the factors that support monosemanticity is significant for pattern interpretation. Although an emerging line of research proposes several attacks to extract and modify information in a monosemantic manner Jermyn et al. (2022) in real cases, the construction of a purely monosemantic model is not feasible due to the unmanageable loss Bricken et al. (2023). To address this problem, another swarm of studies attempts to disentangle the superposition pieces of information (§3.1.2) to achieve a monosemantic understanding. The sparse autoencoder architecture is a promising mechanism for this purpose, mainly via dictionary learning where lexicon features are predefined Sharkey et al. (2022). However, this method has limitations as the actual functioning is strictly correlated to the network structure and the lexicon's sparsity. Bricken et al. (2023) showed that larger autoencoders are able to achieve a finer granularity in feature interpretation and reveal details that cannot be discovered at the low level, i.e., of neurons. These identified features can be used to manipulate the model's output, offering new ways to control and understand models.

### 3.1.2 Superposition

Although different features may activate each neuron, one feature may be distributed over several neurons, while one may be meshed with several features, a phenomenon referred to as overlapping. Elhage et al. (2022) argue that this mechanism results from an imbalance between the number of features and the number of neurons Olah et al. (2020a).

In addition, the superposition enables the representation of additional features. To mitigate interference, it is necessary to introduce non-linear filters Elhage et al. (2022). However, with sparse input features, the superposition effectively supports the representation of these features and allows for calculations such as the absolute value function Elhage et al. (2022). Neurons within models can be monosemantic or polysemantic.

Scherlis et al. (2022) investigates polysemanticity through the lens of feature capacity, indicating the fraction of embedding dimensions consumed by a feature in the representation space. This work indicates that features are represented according to their importance in loss reduction. The most essential features are assigned their dimensions, while the less critical ones may be neglected, and the others share the embedding dimensions. Features end up sharing dimensions only when the assignment of additional capacity does not result in a loss of Scherlis et al. (2022). Furthermore, the relationship between overlap and feature importance has been demonstrated on LLM Gurnee et al. (2023). The experiments show that the first layers tend to represent many overlapping features, whereas the middle layers include neurons dedicated to representing high-level features.

On the other hand, Lecomte et al. (2024) have shown that polysemanticity occurs incidentally due to factors encountered during the training process, such as regularisation and neural noise. In particular, downstream of formal demonstrations in Lecomte et al. (2024), it was shown that a constant fraction of feature collisions, introduced through random initialization, can always result in polysemantic neurons, even when the number of neurons exceeds the number of features.

## 3.2 Attention heads

The in-context learning capabilities of LLMs seem closely related to a strategic part of the Transformers architecture (Brown et al., 2020; Olsson et al., 2022). This special component of attention is called *induction-head*. In particular, induction-heads refer to circuits that complete the pattern by prefix matching and copying previously occurred sequences (Olsson et al., 2022). Their composition makes them fit with sequence matching; in fact, they are structured in two parts. The first part is from the previous layer attending to previous tokens, followed by the current token, which achieves prefix matching and provides the attend-to token (the token following the current token). The second head, i.e., the induction head, copies the attend-to token and increases its output logits. More specifically, this rule means that if models have seen similar patterns such as "[A*][B*]" given current token "[A]", these models are able to predict "[B]" (Olsson et al., 2022). Although this phenomenon is limited to a single token in the toy demonstration, it is possible to demonstrate the correspondence of long prefixes as several consecutive tokens (Chan et al., 2022).

Hence, layers with induction-heads reveal more emergent in-context learning abilities than simple copying activity McDougall et al. (2023). Moreover, several parallel works have been demonstrating the causal relationships between induction heads and in-context learning abilities by observing the change of in-context learning abilities after manipulating induction heads (Olsson et al., 2022; Edelman et al., 2024). Although this theory comprehensively explains the mechanisms behind Transformer with only a few attention layers, further ablation studies are still needed to validate its effectiveness. For this reason, it is important to note that this framework is exclusively based on attention heads and does not incorporate MLP layers Sakarvadia et al. (2023).

## 3.3 Circuits

The circuit is one of the fundamental parts of the mechanistic interpretability field (see Figure **??**). Their origin comes from reverse engineer vision models, in which individual neurons and their connections are viewed as functional units (Olah et al., 2020a). A number of works have shown that features in earlier model levels act as internal units, such as edge detectors. These features are combined through weights to form a circuit unit. This perspective emerges in particular circuits of comprehensible neurons that perform specific functions, such as curve detectors (Cammarata et al., 2020) or frequency detectors (Schubert et al., 2021). Further phenomena, such as symmetrical transformations of basic features, including copying, scaling, flipping, coloring, and rotation, can be achieved with primary neurons known as equivariance or motifs (Olah et al., 2020b).

Although there is extensive literature in computer vision supported by rich insights into vision models, Transformers present new challenges with their unique architecture distinguished by attention blocks. A specific mathematical framework for Transformer circuits has been proposed (Elhage et al., 2021). This framework facilitates the complex architecture of LLM circuits, focusing on decoder-only Transformer models with no more than two layers, all composed entirely of attention blocks. In this toy model, the Transformer incorporates input embedding, residual flow, attention layers, and output embeddings. The attention layers read information from the residual stream and then write their outputs into it. Therefore, communication takes place through read-and-write operations at the layer level Shai et al. (2024).

Each attention head works independently and in parallel, contributing its output to the residual stream Yu & Ananiadou (2024). These components consist of key, query, output, and value vectors, represented as $W_K$, $W_Q$, $W_O$, and $W_V$. There are two types of circuits: i) "query-key" (QK) circuits; ii) "output-value" (OV) circuits (Elhage et al., 2021), as shown in Figure **??**. The QK circuits, formed by $W_Q^T W_K$, are essential in resolving which previously learned token to duplicate information from (Elhage et al., 2021).

It is indispensable for models to recall and retrieve information from before contexts. In contrast, the OV circuits, composed of $W_O$ $W_V$, decide how the current token influences the output logits.

The downstream result shows that Transformers with no layer can model bigram statistics, predicting the next token from the source token. Adding one layer allows the model to capture n-grams patterns. Interestingly, with two layers, transformer models give rise to a concept termed as *induction-head*. These induction heads exist in the second layer and beyond. Usually, they are comprised of heads from their previous layer, which allows suggesting the next token based on the current ones (Elhage et al., 2021).

# 4 Acquiring Knowledge

Mechanistic interpretability hinges on the in-depth study of the structural components of models in order to formalize and manipulate the processes that enable models to acquire Knowledge. In the previous section (§3), we furnished an overview of existing works that have studied the architectural components of artificial neural networks in-depth, with a greater focus on LLMs transformers-based. In this section, we deliver an in-depth analysis of the Knowledge Encoded by LLM representations, including world knowledge and factual Knowledge grasped within these models. In particular, we examine how layer depth and model scale influence this encoding process.

## 4.1 The Seek of Knowledge

Different probing techniques investigate the dynamic structure of representations constructed by LLMs in order to understand whether they encode factual and world knowledge. In particular, probing techniques identify specific directions within the representation space that are essential for understanding certain behaviours and encoding knowledge (Meng et al., 2023a; Zou et al., 2023; Liu et al., 2023). Recent studies have claimed that LLMs are able to learn factual representations of the world and consequently encode them into representations for specific tasks. A series of works Li et al. (2022); Hazineh et al. (2023) have displayed models' ability to track the board state and construct predictions without being explicitly told. Li et al. (2022) uses non-linear probes to reveal world representations within models, specifically in the context of the game of Othello that Hazineh et al. (2023) used to reveal that the analyzed model contains representations that are able to orient decision and habitual processes causally. Later, Nanda et al. (2023b) found that linear representation structures can also perform well in forecasting by simply changing the expression of the card state at each timestamp. The linear and non-linear explanations reveal how models naturally perceive the world, which might differ from humans; moreover, by investigating representations of spatial datasets, Gurnee & Tegmark (2023) exposes the model's proficiency to learn linear representations of space and time across multiple levels.

At the same time, however, these models are able to encode factual knowledge that is more tangible and concrete than the latter. Marks & Tegmark (2023) realize self-corrected binary datasets to study the geometry of representations of true and false information derived from the residual flow of a model. A clear linear structure emerges by applying principal component analysis. The truth directions are exploited to mediate the model's dishonest behaviour locally. In contrast, Burns et al. (2024) reveals that it is possible to discern knowledge from behaviour generated through learning about internal representations.

The manipulation of internal states has also been extensively investigated in Lee et al. (2024); Li et al. (2024b), which have explored toxicity-related vectors within MLP blocks through singular value decomposition, thus proposing effective mitigation techniques and detoxification strategies, as also reported in Bereska & Gavves (2024).

Following the line of research on detoxification and the manifestation of hallucinations, another branch of work aims to extract these undesirable behavioural patterns. In particular, leveraging the direction in the representation space is identified as contributing to a specific behaviour. This directive will then modify the representations that models' behaviours can be controlled (Zou et al., 2023). For example, Li et al. (2024a) employs this technique to probe and improve the honesty of models. Azaria & Mitchell (2023) also successfully distinguishes the truthfulness of statements by simply introducing a classifier on model representations. Recent work has been developed to identify hallucination tokens from the response by

integrating a range of classifiers trained on each layer from separate hidden parts: MLPs and attention layers (CH-Wang et al., 2023).

Function vectors have also been discovered within the attention heads of LLMs, which activate the execution of a specific task across diverse types of inputs. Todd et al. (2023) discovered that these function vectors are shown in different in-context learning tasks and can execute corresponding tasks despite zero-shot inputs. Furthermore, causal interventions at the neuron level can aid in determining the particular neurons encoding spatial coordinates and time facts Gurnee & Tegmark (2023).

## 4.2 Role of Layer Depth and Model Scale

The significance of layer depth and model scale on representations has been an exciting research direction. Formally, research shows that a range of knowledge is well formed down to the intermediate layers. Gurnee & Tegmark (2023) show that spatial and temporal representations reach the best quality up to the middle of the layers in open-source LLMs. Similarly, the function vectors with substantial causal effects are also gathered from the middle layers of LLMs, while the effects are near zero in the more profound layers Todd et al. (2023). Similarly, another study shows that different concepts are well learned in different layers, whereas more straightforward tasks are learned in the early layers. In contrast, complex tasks can only be well learned in the deeper layers Jin et al. (2024a); Ju et al. (2024). However, the underlying reason why the middle layers perform so well remains to be explored. It is stated that more outstanding capabilities are generally acquired as models increase in scale, as discussed in (Wei et al., 2022). Furthermore, spatial and temporal representations are more accurate as the scale of the models increases Gurnee & Tegmark (2023). However, when the scale of the models remains unspecified, the internal mechanism leads to better performance.

## 5 The boundary position: Generalization vs Memorization

The mechanistic interpretability paradigm (§2) is defined by a number of activities that play a key role in understanding the inner dynamics of Artificial Neural Networks (ANN), in this particular case transformers-based Large Language Models (LLMs), from both an architectural (§3) and an internal encoding acquisition (§4) perspective.

In practice, the functionalities studied above are transferred to the study of model training procedures. We then analyze state-of-the-art works that exploit mechanistic interpretability techniques and attempt to define a boundary between generalization and memorization capabilities that merge during training. In particular, we examine two important phenomena: memorization and an exemplification of generalization, i.e., grokking. The latter phenomenon discusses the entanglement between memorization and generalization, as it appears to occur when models suddenly improve validation accuracy beyond overfitting. The analysis of grokking can shed light on how generalization emerges during training. Furthermore, examining memorization, where models are based on statistical patterns rather than causal relationships, can help to discern generalization from memorization's role in model behaviour.

### 5.1 Generalization beyond Memorization

The phenomenon of grokking lies in the fact that models suddenly improve validation accuracy behind extreme overfitting on hyperparametrized artificial neural networks (Power et al., 2022; Nanda et al., 2023a; Murty et al., 2023). Accordingly, the gain in validation accuracy is being interpreted as an income in generalization capability. The empirical reasons for this phenomenon are studied from the perspective of optimization and evaluation algorithms (§5.1.1) and from a data perspective (§5.1.2).

### 5.1.1 Optimization Perspective

The slingshot effect is an indicator of the occurrence of grokking Thilak et al. (2022; 2024); Bhaskar et al. (2024); Bushnaq et al. (2024). During the study of weight norms of the final layers in models that do not use regularization techniques, this effect seems to emerge Huang et al. (2024); Golechha (2024). In particular,

the slingshot effect describes cyclic behaviour during the terminal phase of training, where oscillations between stable and unstable rules occur (training loss spike). The spike co-occurs with a phase where weight norms grow, observed by a phase of norm plateau. Thilak et al. (2022) point out that grokking, non-trivial component adaption, appears only at the beginning of the slingshots effect. The slingshot effect and grokking formation can be modulated by altering the optimizer parameters precisely when operating adaptive optimizers such as Adam (Kingma & Ba, 2014). However, whether this observation holds universally across various scenarios is yet to be determined.

Similarly, Liu et al. (2022b) started to analyze the loss landscapes of ANN. The mismatch between training and test loss landscapes is the cause of grokking, defining it as the LU mechanism. In algorithmic datasets, an L-shaped training loss and a U-shaped test loss reduction concerning weight norms are identified, suggesting an optimal coverage for initializing weight norms Liu et al. (2022b); Furuta et al. (2024). Moreover, this finding only seamlessly transfers to real-world machine learning tasks, where extensive initialization and minor weight decay are continually needed. Earlier works attribute it to a match between the early-phase implicit bias preferring kernel predictors generated by large initialization and a late-phase inferential bias leaning min-norm/margin predictors promoted by minor weight decay Lyu et al. (2023); Mohamadi et al. (2023). Correspondingly, Merrill et al. (2023) suppose this match displays a challenger between a dense subnetwork in the initial step and a spare one behind grokking.

However, from a more model-loss-centered view, as stated in Nakkiran et al. (2021), double descent captures the pattern in which the test accuracy of a model at the log level initially improves, then decreases due to overfitting and finally increases again after gaining the ability to generalize. This effect is evident in the test loss. A unified framework was designed to combine grokking and double descent, treating them as two representations of the same underlying process Davies et al. (2023). The framework attributes the change of generalization to slow pattern learning, further supported by Kumar et al. (2023). Later contributions demonstrate that this transition is displayed both at the level of epochs and patterns Chen et al. (2024a).

### 5.1.2 Data Perspective

On the other side of the coin, although the algorithmic part plays an important role, numerous studies have explored data's role in the learning process. In particular, investigations implemented on two-layer decoder-only transformer-based models have depicted that grokking is closely related to data, representations, and regularization factors. In fact, smaller datasets require additional optimization steps for grokking to happen (Power et al., 2022). In contrast, more samples can decrease the steps required for generalization (Zhu et al., 2024). Liu et al. (2022a) argue that the minimal data needed for grokking depends on the few data points required to learn a robust representation. Moreover, Liu et al. (2022a) have demonstrated that generalization very often coincides with well-structured embeddings. Additionally, regularization actions can rev the onset of grokking, with weight decay particularly effectively strengthening generalization capabilities. Finally, a number of studies are emerging that contradict the previous theories, specifically Zhu et al. (2024); Chen et al. (2024b), which have been demonstrating that massive datasets in LLMs cause grokking to be slightly feasible and that that Transformers can learn implicit reasoning, but only extended training far beyond overfitting Wang et al. (2024).

### 5.2 Memorization

Although generalization is supported by emergent phenomena such as grokking (§5.1), parallel, often introverted series of phenomena occur that models predict with statistical features rather than causal relationships phenomena best exemplified as memorization. The study using slightly corrupted algorithmic datasets with two-layer neural models has demonstrated that memorization can coexist with generalization. Furthermore, memorization can be mitigated by pruning relevant neurons or by regularization (Doshi et al., 2023). Although different regularization methods might not share learning objectives, they all contribute to more reasonable representations. The training process in the analysis consists of two stages: first, there is the grokking process and then the decay of memorization learning (Doshi et al., 2023). However, the underlying causalities behind this process have not been fully comprehended. Similarly, the hypothesis that regularization is the key to this process is under discussion, mainly in light of observing grokking in the absence

of regularization (Kumar et al., 2023). The significance of the rate of feature learning and the number of necessary features is favored in explanations, questioning the role of the weight norm (Kumar et al., 2023).

Nanda et al. (2023a) hypothesizes memorization comprises a step of grokking. The analysis finds that grokking includes three stages: memorization, circuit formation, and memorization cleanup. Moreover, Nanda et al. (2023a) specifies an algorithm that utilizes Discrete Fourier Transforms and trigonometric identities to achieve modular addition by analyzing the model's weights. The circuits enabling this algorithm evolve steadily instead of randomly walking. However, our understanding of the relationship between memorization and grokking still needs to be improved.

### 5.3 Application

The Mechanistic Interpretability paradigm (§2) delivers tools for exploring architectural knowledge (§3), coding, and learning knowledge representations (§4). This understanding enables the analysis of phenomena that emerge during and behind training (§5). Gathering these insights can be harnessed to improve the deep understanding of LLMs mechanisms, improving their efficiency (§5.4), empowering their performance (§5.5), and better aligning them with human values and preferences (§5.6) by reducing the ongoing gap between humans and models.

### 5.4 Improving Efficiency

Attention heads and neuron activations play a fundamental role in Transformers' architecture Neo et al. (2024). Hence, causal tracing and analysis of causal mediation from mechanistic interpretability is a fundamental technique for deep understanding of models. Stolfo et al. (2023); Hou et al. (2023) study the importance of the attention mechanism in revealing how the model processes input, showing that this mechanism enables models to extract query information in the final token in the first levels of transformers-based models. In addition, result information is incorporated into the residual stream in the last MLP levels. Localizing this information improves fine-tuning on specific tasks Prakash et al. (2024) and allows one to focus on only a part of the models while ignoring information that is superfluous to the specific task (pruning technique) Hu et al. (2021); Wu et al. (2024a); Held & Yang (2023).

However, these studies are merely specializations of the study proposed in (Jain et al., 2023) that examines the differences between the pre-training and fine-tuning phases with mechanistic interpretability tools. Jain et al. (2023) show that fine-tuning retains all the skills learned in the pre-training phase. Transformations between pre-training and fine-tuning result from wrappers located in the MLPs learned at the top of the models. In particular, wrappers can be eliminated by pruning some neurons or retraining an unrelated downstream task. This finding sheds light on potential security problems associated with current alignment approaches.

By exploiting these attacks, representation engineering aims to directly manipulate representations without needing optimization or further labeled data Zou et al. (2023). This technique has proven effective specifically as a specialization of model pruning. Wu et al. (2024a) propose techniques for fine-tuning models with representation engineering and performing similar and even more reasonable performance than more evolved fine-tuning methods as in Turner et al. (2023); Wu et al. (2024b). After that, Wu et al. (2024a) reveals the feasibility of fine-tuning models via editing representations. Unlike conventional parameter-efficient fine-tuning, representation editing focuses on learning more trainable parameters to change representations directly other than models' parameters. The trainable parameters have been reduced to a factor of 32 reached to that of LoRA (Hu et al., 2021). Geiger et al. (2024) employs distributed alignment tracking to find a set of linear subspace implementing interventions. This strategy outperforms most PEFT models on different scenarios (Wu et al., 2024b; Hase et al., 2023).

### 5.5 Improving Performances

As well as efficiency, tangible results were also observed in terms of performance. Cao et al. (2021) demonstrated that it is feasible to edit factual knowledge by changing the weights of specific neurons in MLPs. Meng et al. (2022) adopts this approach by changing neural computations connected to recall factual knowl-

edge. Afterward, they expand this method to allow multiple edits simultaneously (Meng et al., 2023b). Although these methods are effective for targeted edits, their ability to edit relevant knowledge and control forgetting still requires further research (Cohen et al., 2023).

Stoehr et al. (2024) suggests that the sections memorized by a model can be pinpointed using high-gradient weights in the attention heads of the lower layers. This research employs localization techniques to identify detailed attention heads, which are fine-tuned to unlearn the memorized knowledge. This approach swears to improve privacy protection in LLMs, although an exhaustive evaluation is yet required Bereska & Gavves (2024).

In particular, facts are encoded in the representation space, making assembling representations a natural contender for editing models' outputs. So far, most analyses focus on modifying representations at inference time, while the influence of permanent modifications has yet to be studied. Recent work provides a more precise way to edit model representations to change their output distributions (Hernandez et al., 2023; Xu et al., 2024). Rather of only counting the derived vectors into effects representations, this investigation directly adjusts the embedding of a related entity to trigger targeted outputs. Therefore, the modified entity's position in the embedding space has changed, leading to a causal effect on model generations.

### 5.6 Mechanistic Intepreptability to refine Models' capabilities

From a mechanistic point of view, different practical applications have been proposed to improve and refine human model alignment. Particularly in the case of bias, where current measures are based on probing, designed prompts, known as prompt engineering, are primarily exploited. The completeness of these prompts determines the effectiveness of these measures. However, prompts can only capture recognized biases using a finite set of examples often confined to specific tasks. In this way, biases that have been learned but have yet to be explicitly known and made explicit across generations cannot be detected.

To address these issues, several novel works have been approaching the problem from a mechanistic point of view Zhang & Nanda (2024); Sharma et al. (2024); Rajendran et al. (2024); Tamkin et al. (2023). Zhang & Nanda (2024) deligned the guidelines previously used Campbell et al. (2023) to locating the attention heads responsible for lying with a linear survey and an activation patch. Yang et al. (2023) focusing on stereotype recognition estimate bias scores of attention heads in pre-trained LLMs. They implemented a method to ensure the accuracy of determining biased heads by comparing the changes in attention scores between biased and regular heads.

Representation engineering has emerged as a promising avenue for detecting biases within embedding space. Geva et al. (2022b) and later Sharma et al. (2024) suggest that MLPs operate on token representations to alter the distribution of output vocabulary. After reverse engineering, the output from each feed-forward layer can be seen as sub-updates to output vocabulary distributions, essentially promoting certain high-level concepts that could effectively mitigate toxicity levels in LLMs. Lee et al. (2024) identified multiple representation vectors within MLPs that encourage models' undesired behaviours. Hence, they decomposed the vectors using singular value decomposition, allowing them to pinpoint specific dimensions contributing to toxicity. Finally, Jin et al. (2024b) interpreted the mechanism of knowledge conflicts through the lens of information flow and mitigating conflicts by precise systematic interventions.

## 6 Final Discussion & Future Challenges

This paper explores the innovative paradigms that are taking the next step of explainability: mechanistic interpretability and representation engineering. We deliver a systematic overview of the architectural composition of knowledge within Large Language Models (LLMs) and the internal representations. We analyzed several dynamics underlying the investigation of phenomena that represent the borderline between memorization and generalization. We concluded by examining how insights from these research lines can improve the performance of LLMs through model editing, improve efficiency via pruning, and better align models to human preferences. There are some preliminary progress in discovering the inner workings of LLMs, but looking further, multiple challenges have emerged. Although LLMs have encoded much real-world knowledge, current research has revealed only a tiny part of the encoded knowledge. Hence, future efforts should

develop scalable techniques to effectively analyze and interpret the intricate knowledge structures embedded in models. LLMs demonstrated remarkable reasoning capabilities by displaying human-like cognitive capacities. However, the current understanding of how high-level reasoning capabilities emerge from the interaction of architectural components and training dynamics needs to be improved. More work is needed to unravel the intricate mechanisms that underpin these advanced reasoning capabilities. In summary, the insights gleaned from mechanistic interpretability and representation engineering have laid the foundation for initial strides in editing, pruning, and model alignment. However, the progress made thus far has been relatively modest. It is therefore of utmost importance that we continue to build on these insights and develop techniques that can significantly enhance the performance of LLMs.

## 7 Conclusions

The internal mechanisms that enable Large Language Models (LLMs) generalization and reasoning capabilities must be fully understood. This paper explored the current state of explainability approaches related to investigating emergent mechanisms within LLMs. In particular, using a mechanistic interpretability viewpoint, we explored the strategic components expected to lay the foundations for these capabilities. Then, we investigated the means of quantifying the knowledge acquired and provided by LLMs by discerning the composition and the encoding mechanisms of knowledge within parameters by taking different points related to the mechanistic interpretability and representation engineering field. Hence, we observed the state of applications using a mechanistic perspective that allowed us to explain emergent phenomena that occur in training dynamics, such as grokking, which unveil the generalization capabilities of LLMs. Finally, we examined how insights from these analyses can improve the performance of LLMs via model modification, improve efficiency, and better align models with human preferences. We conclude our contribution by pointing out that although emerging insights gained from mechanistic interpretability are enabling preliminary efforts in areas such as editing, pruning, and model alignment, further work is needed to exploit these insights and deliver techniques to improve the LLMs' performances.

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
