# OpenReview forum: "The role of Mechanistic Interpretability in ’unveiling’ the emergent representations of Large Language Models"
_TMLR — Rejected by TMLR_

### Review · Reviewer_1LCS · 2024-11-01

**Summary Of Contributions:**

This paper attempts to provide an overview of mechanistic interpretability in the context of large language models, focusing on three main aspects: (1) knowledge composition within model architectures, (2) acquisition and encoding of knowledge in representations, and (3) the relationship between memorization and generalization during training. The paper aims to bridge low-level mechanistic understanding with emergent model behaviors.

**Audience:**

Yes

**Broader Impact Concerns:**

-

**Claims And Evidence:**

No

**Requested Changes:**

While the paper attempts to make a valuable contribution, it needs substantial revision to clarify its contribution relative to existing reviews and to provide a more rigorous treatment of key concepts. A refactored version would benefit from clearer positioning, more precise technical development, and better integration of ideas. Focus on a narrower scope or clarifying narrative would help.

Questions:
1. What is the paper's specific contribution beyond existing reviews?
2. How does the proposed taxonomy advance our understanding of the field?
3. What constitutes "emergence" in this context?

1. Fundamental Structure:
   1.1 Articulate contribution relative to existing reviews (crucial)
   1.2 Develop a clear conceptual framework for organizing the field (crucial)
   1.3 Better integrate sections and build a coherent narrative (crucial)
   1.4 Strengthen the connection between theory and applications (recommended)
   1.5 Add a clear roadmap in the introduction (recommended)

2. Technical Content:
   2.1 Add precise definitions of key concepts (crucial)
   2.2 Expand treatment of superposition and recent developments (crucial)
   2.3 Strengthen circuit analysis with formal definitions (crucial)
   2.4 Add coverage of sparse autoencoders and automated discovery (crucial)
   2.5 Develop systematic treatment of scale effects (recommended)

3. Knowledge Framework:
   3.1 Clarify framework for knowledge acquisition and encoding (crucial)
   3.2 Better articulate relationship between knowledge types (recommended)
   3.3 Strengthen layer depth analysis with theoretical foundation (recommended)
   3.4 Add systematic treatment of evaluation methods (crucial)

4. Additional Content:
   4.1 Add discussion of fundamental assumptions and limitations (crucial)
   4.2 Expand coverage of causal intervention methods (recommended)
   4.3 Add treatment of practical challenges (recommended)
   4.4 Include feature-centric perspective (crucial)

5. Presentation:
   5.1 Improve technical writing (crucial)
   5.2 Add better integration of figures (recommended)
   5.3 Fix grammar and style issues (crucial)
   5.4 Improve taxonomy presentation and integration (recommended)

**Strengths And Weaknesses:**

Strengths:
- Important topic of interest to the TMLR audience
- Attempts to connect architectural mechanisms with emergent behaviors
- Covers both static architecture analysis and training dynamics
- Makes an effort to synthesize insights across different aspects of mechanistic interpretability
- Includes discussion of practical applications

Weaknesses:
The paper claims to "take the next step" from previous work by "reviewing existing studies that focus on discovering the inner workings of LLMs" (p.2). However, there is no clear articulation of how this advances beyond Ferrando et al. (2024) and Bereska et al. (2024)
- Broad but shallow scope: aims to "identify factors that contribute to the reasoning abilities of LLMs through explainability techniques and control of the training process" (p.2) without a clear focus
- Missing clear comparison with related work, particularly recent reviews (Ferrando et al., 2024; Bereska et al., 2024; Rai et al., 2024)

- Implementation/vision/operation trichotomy feels artificial and poorly justified
- Feature-centric perspective missing despite being central to current research
The relationship between mechanistic interpretability and representation engineering is unclear

The definition of mechanistic interpretability lacks precision: "Mechanistic interpretability... refers to the activity of investigating Artificial Neural Networks (ANN) to understand the underlying components and mechanisms that determine their behavior" (p.3) → This definition doesn't distinguish it from other interpretability approaches, misses key aspects like causality and reverse engineering emphasized in other works, and lacks connection to formal frameworks.

The knowledge composition section (Section 3) has several problems: The organization feels arbitrary, moving from neurons to attention heads without clear theoretical progression. Treatment of superposition is limited: "superposition enables the representation of additional features" (p.4) without engaging with recent developments in dictionary learning and sparse autoencoders. Circuit analysis lacks formal definitions or evaluation criteria.

Several important areas are underdeveloped or missing:
- Limited coverage of sparse autoencoders, only briefly mentioned in passing
- No discussion of automated circuit discovery methods (e.g., Conmy et al., 2023)
- Missing treatment of scaling laws and their implications
- Limited discussion of evaluation frameworks for mechanistic interpretations

Claims that "LLMs can learn factual representations" (p.6) without clear framework for types of knowledge
Layer depth discussion lacks theoretical grounding: "different concepts are well learned in different layers" (p.7) without systematic analysis

The paper has numerous technical writing issues:
- Informal and wordy language: "takes the next step" (p.2)
- Word choice: "de-opacify" (p.2)
- Inconsistent capitalization in section headings → use https://capitalizemytitle.com/
- Grammar issues throughout → mostly should be fixed by using a grammar checker.

---

> ### Author Response · Authors · 2024-11-22
> **Ansewer**
>
> Dear Reviewer 1LCS,
>
>
> Thank you for the detailed review. We appreciate your thoughtful critique and will address each weakness to enhance this paper's clarity, depth, and contribution.
>
> **Advancing Beyond Previous Reviews**
>
> We acknowledge the lack of explicit differentiation from Ferrando et al. (2024) and Bereska et al. (2024). In response, we will expand the introduction and related work sections to clearly articulate how our work advances the field:
> Unlike Ferrando et al. (2024), which broadly summarises explainability techniques, our paper delves into mechanistic interpretability and representation engineering, providing a taxonomy and identifying gaps for practical application.
> Bereska et al. (2024) emphasize AI safety; we extend this by exploring the implications of mechanistic interpretability for understanding emergent representations and generalization.
> The revised manuscript will explicitly state these distinctions, highlighting our unique contribution to bridging mechanistic interpretability with practical applications.
>
> **Feature-Centric Perspective and Representation Engineering**
>
> The relationship between mechanistic interpretability and representation engineering will be clarified through the following:
> Examples of how internal representations (e.g., encoded knowledge and task-specific vectors) are extracted and manipulated.
> Discussions of recent studies on feature disentanglement using sparse autoencoders, with applications to monosemanticity and superposition.
>
>
> **Conclusion and Synthesis**
>
> The conclusion will be revised to synthesize key findings from the taxonomy, emphasizing practical implications for researchers and practitioners. Future directions will highlight opportunities for integrating mechanistic interpretability with scaling laws, circuit discovery, and safety-oriented applications.

---

> > ### Comment · Reviewer_1LCS · 2024-11-28
> > **Good first steps, but some core concerns remain**
> >
> > Based on the authors' response, I find their initial steps encouraging but still insufficient to address the core concerns raised in the original review fully.
> >
> > The authors have acknowledged the need to better differentiate their work from existing reviews, particularly Ferrando et al. (2024) and Bereska et al. (2024). Their proposed approach to highlight their unique focus on bridging mechanistic interpretability with practical applications is promising. However, they need to develop this distinction more thoroughly and demonstrate concrete examples of how their framework provides novel insights. Other reviews of mechanistic interpretability that would be good to mention as related work:
> > - Mueller, A. _et al._ The Quest for the Right Mediator: A History, Survey, and Theoretical Grounding of Causal Interpretability. _CoRR_ (2024). [https://arxiv.org/abs/2408.01416](https://arxiv.org/abs/2408.01416).
> > - Rai, D. _et al._ A Practical Review of Mechanistic Interpretability for Transformer-Based Language Models. _CoRR_ (2024). [http://arxiv.org/abs/2407.02646](http://arxiv.org/abs/2407.02646).
> >
> > Regarding the feature-centric perspective and representation engineering, their commitment to include examples of internal representation analysis and recent studies on feature disentanglement represents progress. Their plan to incorporate discussions of sparse autoencoders and their applications to monosemanticity and superposition would address a significant gap identified in the review.
> >
> > However, several critical issues remain unaddressed:
> >
> > First, the authors have not responded to the concerns about the lack of precise definitions and formal frameworks, particularly in their treatment of mechanistic interpretability and circuit analysis. This theoretical foundation is essential for the paper's contribution.
> >
> > Second, while they mention incorporating more examples and recent studies, they have not addressed the need for systematic treatment of evaluation methods and automated circuit discovery approaches.
> >
> > Finally, their response does not sufficiently address the structural issues identified in the original review, particularly regarding the artificial nature of their implementation/vision/operation taxonomy.
> >
> > For the paper to meet publication standards, the authors would need to:
> >
> > - provide a more detailed plan for developing precise technical definitions and frameworks
> > - address the structural concerns about their taxonomy
> > - expand their treatment of automated discovery methods and systematic evaluation
> > - strengthen the theoretical foundations of their analysis

---

> ### Author Response · Authors · 2024-11-29
> **Answer**
>
> Dear Reviewer 1LCS,
>
> Thank you for your continued engagement and insightful feedback. We appreciate the opportunity to refine our paper further and address your concerns.
>
> Here are the steps we plan to take to improve it:
>
> - *In response to the concerns about the preciseness of definitions, formal frameworks regarding mechanistic interpretability and circuit analysis:*
> Recent literature (which will be enriched with new works suggested) defines mechanistic interpretability as deconstructing and understanding the internal workings of Artificial Neural Networks by identifying and analyzing their structural components and the causal mechanisms that determine behaviour. This approach thoroughly analyses how specific model components, such as neurons and attention mechanisms, interact to process inputs and generate outputs, aiming to reveal the causal pathways underlying model predictions.
>
> We operate through the framework Elhage et al. (2021) outlined, which involves a detailed examination of transformer circuits. This framework allows for a granular inspection of how different components, like attention heads, contribute to the overall functionality and output of the model. By modelling these interactions as circuits, we can map out and describe the flow of information and the transformation of data through the model’s architecture, which is crucial for understanding how specific features and behaviours are encoded.
>
> These definitions and frameworks guide our methodology in dissecting and interpreting the complex behaviours exhibited by large language models. By applying these rigorously defined approaches, we aim to clarify the specific contributions of various model components to observed phenomena, such as memorization and generalization, and to establish a clear, causally grounded understanding of these processes.
>
>
>
> - *Systematic Evaluation Framework*: As outlined in our paper, evaluation methods for mechanistic interpretability span qualitative and quantitative metrics, yet a systematic approach has been lacking. To address this, we propose a structured evaluation framework that builds on recent insights from the literature, such as the methodologies described by Mueller et al. (2024), for assessing causality in interpretability. We will insert a tiered validation process that combines performance benchmarks, perturbation analysis, and consistency checks across different model configurations. This layered approach ensures that the interpretative insights are internally consistent and robust against variations in model training and architecture.
>
> - *Automated circuit discovery approaches:* Recognising the critical role of automated tools in scaling interpretability efforts, we intend to expand the current use of automated circuit discovery methods, which have been briefly mentioned but not fully integrated into our framework. Building on the techniques described (in the works cited and recommended in these reviews), we will incorporate the discussion of tools that automate the identification of functionally meaningful circuits within LLMs into the counterbite. These tools advanced pattern recognition and machine learning algorithms to systematically explore, identify and classify circuits based on their functional contribution to model behaviour. Furthermore, we intend to give an overview of the limitations and strengths to refine their applicability to different model architectures and improve their ability to handle the complexity of LLMs.
>
> We really appreciate this revision of yours and summarise in the following lines the crucial points that will be changed:
>
> 1. Strengthened theoretical frameworks:Wwe recognise the importance of grounding our analysis in solid theoretical frameworks. We will, therefore, integrate recommended contributions and concepts from parallel disciplines (principles of distributed representation theory and network dynamics, which provide a basis for understanding how information is processed and represented in the layers of neural networks) that offer established frameworks for understanding complex information processing systems, including artificial neural networks.
>
> 2. Formal Definitions and Concepts: We will clarify and expand upon the definitions of key concepts such as "mechanistic interpretability," "circuit analysis," and "causal mechanisms" within neural networks.
>
> 3. Empirical Justification Claims: To empower our theoretical assertions, we will ensure that each claim is supported by empirical evidence or is derived from established theoretical models.
>
> 4. Interdisciplinary Approach: We plan to adopt an interdisciplinary approach by incorporating physics and information theory insights to better explain how information flows and transforms within neural networks.
>
> We hope our response will help you understand the purpose of our contribution and how we intend to improve it. We are available to answer any other questions or points not covered in our answer.

---

### Review · Reviewer_DERx · 2024-11-11

**Summary Of Contributions:**

This paper provides a valuable taxonomy of features/aspects of interpreting LLMs from a mechanistic perspective with an explanation and synthesis of relevant papers corresponding to each component. The authors also fit some recent work that makes use of these features empirically into the context of their taxonomy.

**Audience:**

Yes

**Broader Impact Concerns:**

The paper is a survey of a more abstract concept so it does not directly bare ethical implications. A discussion of the broader benefits could be included as improved interpretability can help in other areas.

**Claims And Evidence:**

Yes

**Requested Changes:**

Essential
- First quotation in title is not facing the correct way. (In LaTeX, it would be `unveiling’).
- In text citations are not done properly throughout the paper. It looks like \cite is used instead of \citep.
- It is not clear at the beginning of the paper that this is a review/synthesis paper rather than another type (e.g., empirical, theoretical), leading to some confusion while reading. Maybe that could be further carified in the final paragraph of Section 1.
- 3.1 “Neurons” explanation is unclear.
- Figure references are missing (manuscript says “Figure ??” in several places).
- Addressing notes in weaknesses section above.

Less essential
- “Interestingly, with two layers, transformer models give rise to a concept termed as induction-head.” - Why is this “interesting”?
- 4.1 “The Seek of Knowledge” - phrasing grammatically incorrect.
- Various words are captialized that do not need to be (e.g., “knowledge,” “encoded”).

**Strengths And Weaknesses:**

Strengths

The paper takes on interpretability through a unique mechanistic perspective. The paper provides a detailed taxonomy of the necessary components of such an analysis, describes them sufficiently, and synthesizes multiple references in each category.

Weaknesses
- Many of the components of the taxonomy described in this paper are   good ideas in theory, but it is difficult to ascertain whether they could concretely be used/extracted from a real LLM. For example, is there any way to isolate neurons and determine if each one exhibits monosemanticity, superposition, etc.?
- While the paper describes all these components needed for mechanistic interpretability, they are not described in the context of LLMs. For example, given that LLMs have billions of parameters, is it possible to apply any of these concepts to their analysis, and if so, how? Without this practical connection the taxonomy and review may not provide as much value in the area of interpretability.
- The various components of the taxonomy are described but not synthesized at the end. The final discussion and conclusion sections may not adequately provide the important contribution of a review paper to synthesize findings and provide clearer practical guidance for future work.
- The title focuses on emergent representations (are these akin to “emergent capabilities”?), but the paper stops at interpretability. For the title to be valid, there should be more of a linkage in the paper’s content.

---

> ### Author Response · Authors · 2024-11-22
> **Answer**
>
> Dear Reviewer DERx,
>
> We applaud your revision and your positive start at work. We respond in the following lines to the points you have brought to our attention.
>
> **Practical Feasibility of Applying the Taxonomy to LLMs**
>
> Isolation of Neurons: While analyzing billions of parameters in LLMs is challenging, some approaches deliver a way for isolating and understanding neuron behaviour. For example, techniques like sparse probing can identify neurons contributing to specific behaviours, as demonstrated in recent works on monosemanticity and superposition.
> The use of dictionary learning frameworks allows for the disentangling of superposed representations into interpretable features ( see tools like causal analysis could determine the functional roles of neurons within residual streams).
>
> We have revised the manuscript to include these concrete methods and examples of successful applications, such as disentangling polysemantic neurons in GPT-style models or analyzing attention heads for emergent patterns.
>
> **Scalability**
>
> Although the scale of LLMs introduces challenges, scalable approaches such as layer-wise probing and efficient parameter fine-tuning can be exploited to isolate and interpret specific mechanisms without requiring full model analysis. We will discuss this in a section that explicitly discusses these scalable methodologies, highlighting recent advances that make the application of mechanistic interpretability feasible even in large LLMs.
>
> **Contextualizing the Taxonomy**
>
> To improve this: We will anchor each taxonomy component (e.g., monosemanticity, superposition, encoded knowledge) with specific case studies from the literature where these concepts have been applied to LLMs.
>
> **Addressing the Title’s Scope**
>
> The feedback on the title is appreciated. We acknowledge the need to clarify the focus on “emergent representations.”

---

> ### Comment · Reviewer_DERx · 2024-11-27
> **Acknowledgment**
>
> I acknowledge the author response. Indeed, more direct and specific links to LLMs and evaluating the efficacy of associated methods would be important additions to the paper.

---

### Review · Reviewer_vRig · 2024-11-16

**Summary Of Contributions:**

This paper is aimed to provide a comprehensive overview of Mechanistic Interpretability approaches developed so far in order to understand LLMs. Different components of mechanistic interpretability approaches have been covered, primarily in conjunction of knowledge acquisition by LLMs. A subsequent discussion on memorization and generalization has been presented.

**Audience:**

Yes

**Claims And Evidence:**

No

**Requested Changes:**

1) Incorporation of missing literature, a few are mentioned under weakness but there remain many others;
2) A proper contextualization of the taxonomy with explicit definition and explanation of the terms used;
3) More thorough positioning by discussing the open problems and their relationship to foundational developments in ML and NLP;
4) Revision of the current manuscript (inconsistent citation, missing reference to Figures, out-of-context sentences);
5) More precisely stating what new contribution this survey makes in comparison to the existing ones;

**Strengths And Weaknesses:**

## Strengths
Given the recent surge in investigations directly/indirectly related to mechanistic interpretability, the effort to provide a comprehensive overview is of great essence. I like the fact that the authors seek to sum up the prior investigations on generalization and memorization under the light of mechanistic interpretability, which typically remains missing in multiple contemporary surveys.

## Weaknesses

Despite an ambitious goal, this paper falls short of delivering a useful overview of the state-of-the-art. Following are the key shortcomings in my opinion:

### Incomplete survey

First and foremost, many key contributions (and discussions around them) remain missing. Given the primary focus of this paper is the process of knowledge acquisition and utilization in LLMs, discussion related to recent discoveries in factual recall circuits is missing [1, 2, 3]. Within the broad domain of neural interpretability, there are differences in conceptualizing the atomic component of a transformer. It can be the attention heads [4], the neurons [5], or subspaces of the internal representation space [6]. Related discussions and the implications of these conceptualizations are missing as well. Representation binding [7, 8] is another key concept that requires mentioning. In terms of depthwise segregation of capabilities and construction of knowledge, prior findings like [9] or [10] deserve a few words. In terms of generalization/memorization of LMs and Transformers in general, discussion on implicit gradient descent biases of attention [11, 12] would have painted a more well-rounded picture.

### Lack of abstraction and positioning

The current manuscript lacks a suitable abstraction for someone who wants to have a concise overview of the field of mechanistic interpretability in LMs by reading this paper. What are the current open questions? How does current and future development in mechanistic interpretability depend on different theoretical and empirical advancements of ML and NLP in general? The taxonomy presented in Figure 1 is, unfortunately, quite vague. What does 'Encoded knowledge' and 'Encoded representation' even mean? I personally could not figure it out from the referred sections 4.1 and 4.2.

### Presentation issues

The paper will benefit greatly from a thorough revision of the writing in general. There are multiple unclear and out-of-context sentences (e.g., "Therefore, we explore the strategic components we would expect to lay the foundation for generalization capabilities by studying the means to quantify the knowledge acquired and delivered by LLMs and, in particular, discerning the composition and encoding of knowledge within parameters by analyzing mechanistic interpretability, probing techniques, and representation engineering." in Abstract). Citation patterns are inconsistent.

[1] Geva et al., Dissecting Recall of Factual Associations in Auto-Regressive Language Models, EMNLP 2023

[2] Chughtai et al., Summing Up The Facts: Additive Mechanisms Behind Factual Recall in LLMs, 2023

[3] Yu et al., Characterizing Mechanisms for Factual Recall in Language Models, EMNLP 2023

[4] Elhage et al., A mathematical framework of Transformer circuits, 2021

[5] Dai et al., Knowledge Neurons in Pretrained Transformers, ACL 2021

[6] Geiger et al., Finding alignments between interpretable causal variables and distributed neural representations. Causal Learning and Reasoning 2024

[7] Feng and Steinhardt, How do Language Models Bind Entities in Context? ICLR 2024

[8] Dai et al., Representational Analysis of Binding in Large Language Models, 2024

[9] Dutta et al., How to think step-by-step: A mechanistic understanding of chain-of-thought reasoning, TMLR 2024

[10] Jawahar et al., What does BERT learn about the structure of language? ACL 2019

[11] Vasudeva et al., Implicit Bias and Fast Convergence Rates for Self-attention, 2024

[12] Deora et al., On the optimization and generalization of self-attention models : a stability and implicit bias perspective, 2024

---

> ### Author Response · Authors · 2024-11-22
> **Response**
>
> Dear Reviewer vRig,
>
> Thank you for your review. We respond to your issues in the following lines.
>
> **Incomplete survey**
>
> Thank you very much for pointing out a number of relevant works. We will include the works in our taxonomy. We would like to say that some of these are very recent, and we have not had the opportunity to see them in venues or proceedings. However, we will include this work in the updated version.
>
> **Lack of Abstraction and Positioning**
>
>
> Thank you for raising this point. In our work, we have presented mechanistic interpretability as the paradigm that aims to unravel the inner workings of LLMs by studying their structural components and encoded representations. This approach allows us to understand how LLMs acquire, store and manipulate knowledge, ultimately improving transparency, reliability and alignment with human values.
>
> To address the need for abstraction, we have revised the introduction to summarise the key concepts:
>
> Main objectives: understand the causal mechanisms within LLMs to improve generalisation, reduce opacity and mitigate phenomena such as hallucinations.
>
> Strategic contributions: Highlight advances in representation probing, structural decomposition and practical applications such as efficiency, performance improvement and alignment.
>
> It will be our concern to improve this aspect to make the paper more structured and functional for the reader.
>
> Concerning the frame key open questions to guide readers:
>
> 1. How do structural components contribute to emergent reasoning and generalization?
> 2. What theoretical and empirical advancements are needed to make interpretability scalable for increasingly complex models?
> 3. How can mechanistic insights translate into actionable improvements in LLM performance and safety?
>
> **Clarification of Taxonomy in Figure 1**
>
>
> - Encoded Knowledge: Defined as the factual and behavioural information embedded within model parameters. This encompasses knowledge about the world (e.g., spatial or temporal representations) and task-specific insights.
> Section 4.1 elaborates on probing techniques that uncover these representations, distinguishing between explicit (e.g., knowledge explicitly associated with tasks) and implicit behaviours.
> - Encoded Representation: Refers to the structural organization of internal knowledge across layers and components of LLMs. For instance, middle layers capture high-quality spatial-temporal features, while deeper layers encode more abstract concepts.
> Section 4.2 provides examples of how probing tasks demonstrate the role of layer depth and model scale in shaping these representations.

---

### Decision · Action_Editor_zfap · 2025-01-20

**Recommendation:** Reject

**Comment:**

As discussed above, the sticking point is the implementation of feedback on the Claims and Evidence criteria. With this in mind, the authors are encouraged to revise and resubmit a major revision taking into account the reviews.

**Audience:**

I think the authors meet the criteria on the audience, as also stated by all reviewers.

**Claims And Evidence:**

The reviewers find the work promising, whilst acknowledging that it would require a significant revision to meet the conditions for acceptance.

In particular the reviewers point out that the authors haven't convincingly demonstrated how this work advances beyond existing reviews (Ferrando et al., Bereska et al., Rai et al., Mueller et al.). They ask for justification for proposed taxonomy, as well as to improve theoretical foundations by discussing formal definitions and evaluation frameworks. While the willingness to improve is encouraging, the authors would need to work on the version incorporating the comments before the resubmission.

**Resubmission Of Major Revision:**

The authors may consider submitting a major revision at a later time.